# Diagnostic Performance of Kwak, EU, ACR, and Korean TIRADS as Well as ATA Guidelines for the Ultrasound Risk Stratification of Non-Autonomously Functioning Thyroid Nodules in a Region with Long History of Iodine Deficiency: A German Multicenter Trial

**DOI:** 10.3390/cancers13174467

**Published:** 2021-09-04

**Authors:** Philipp Seifert, Simone Schenke, Michael Zimny, Alexander Stahl, Michael Grunert, Burkhard Klemenz, Martin Freesmeyer, Michael C. Kreissl, Ken Herrmann, Rainer Görges

**Affiliations:** 1Clinic of Nuclear Medicine, Jena University Hospital, 07749 Jena, Germany; martin.freesmeyer@med.uni-jena.de; 2Division of Nuclear Medicine, Department of Radiology and Nuclear Medicine, Magdeburg University Hospital, 39120 Magdeburg, Germany; michael.kreissl@med.ovgu.de; 3Institute for Nuclear Medicine Hanau, 63450 Giessen, Germany; zimny@nuklearmedizin-hanau.de; 4Institute for Radiology and Nuclear Medicine RIZ, 86150 Augsburg, Germany; dr.alexander.stahl@gmx.de; 5Department of Nuclear Medicine, German Armed Forces Hospital of Ulm, 89081 Ulm, Germany; michael.grunert@uni-ulm.de (M.G.); burkhard.klemenz@uni-ulm.de (B.K.); 6Department of Nuclear Medicine, Essen University Hospital, 45147 Essen, Germany; ken.herrmann@uk-essen.de (K.H.); rainer.goerges@uni-due.de (R.G.); 7Joint Practice for Nuclear Medicine, Duisburg (Moers), 47441 Duisburg, Germany

**Keywords:** thyroid, cancer, nodule, ultrasound, scintigraphy, non-autonomously functioning, thyroid imaging reporting and data systems (TIRADS), risk of malignancy (ROM)

## Abstract

**Simple Summary:**

In Germany, thyroid nodules can be detected by ultrasound examinations in over 30% of the adult population, mainly as a result of prolonged nutritive iodine deficiency. Although only a small proportion of the nodules are malignant, it is important to have a reliable examination method that not only can detect these few thyroid carcinomas with a high degree of certainty, but also not be unnecessarily invasive for the much larger number of benign nodules. Ultrasound is the method of choice, and ultrasound-based risk stratification systems are important tools in clinical care. However, many different systems have been introduced within the last decade. The aim of this study was to evaluate five common ultrasound risk stratification systems for their diagnostic accuracy of thyroid nodules from an area with long history of iodine deficiency.

**Abstract:**

Germany has a long history of insufficient iodine supply and thyroid nodules occur in over 30% of the adult population, the vast majority of which are benign. Non-invasive diagnostics remain challenging, and ultrasound-based risk stratification systems are essential for selecting lesions requiring further clarification. However, no recommendation can yet be made about which system performs the best for iodine deficiency areas. In a German multicenter approach, 1211 thyroid nodules from 849 consecutive patients with cytological or histopathological results were enrolled. Scintigraphically hyperfunctioning lesions were excluded. Ultrasound features were prospectively recorded, and the resulting classifications according to five risk stratification systems were retrospectively determined. Observations determined 1022 benign and 189 malignant lesions. The diagnostic accuracies were 0.79, 0.78, 0.70, 0.82, and 0.79 for Kwak Thyroid Imaging Reporting and Data System (Kwak-TIRADS), American College of Radiology (ACR) TI-RADS, European Thyroid Association (EU)-TIRADS, Korean-TIRADS, and American Thyroid Association (ATA) Guidelines, respectively. Receiver Operating Curves revealed Areas under the Curve of 0.803, 0.795, 0.800, 0.805, and 0.801, respectively. According to the ATA Guidelines, 135 thyroid nodules (11.1%) could not be classified. Kwak-TIRADS, ACR TI-RADS, and Korean-TIRADS outperformed EU-TIRADS and ATA Guidelines and therefore can be primarily recommended for non-autonomously functioning lesions in areas with a history of iodine deficiency.

## 1. Introduction

Iodine deficiency is a well-known risk factor in the development of nodular thyroid disease [1]. Although nutritive iodine supply in the German population has improved in the recent years, Germany has a long history of iodine deficiency and the requirements of the World Health Organization (WHO) have not yet been fully met [2,3,4,5]. The prevalence of thyroid nodules (TNs) ranges from 12.5% in young men to over 80% in older women [6,7,8,9]. Since the vast majority of the detected TNs are benign, the diagnostic challenge is to reliably detect malignant nodules while avoiding unnecessary interventions for benign lesions [10].

Thyroid ultrasound (US) is a non-invasive, cost-effective, and accurate method for detecting and describing TNs [11]. It is also the method of choice for assessing and selecting TNs for further diagnostic procedures such as fine-needle cytology (FNC) to rule-out malignancy [12,13,14]. During the last decade, several international societies have published different US-based risk stratification systems (RSSs, Thyroid Imaging Reporting and Data System, TIRADS) based on US features and lesion size. The aim was to improve diagnostic performance of thyroid US, to reduce unnecessary interventions, and to provide a standardized terminology for physicians [12,13,15,16,17,18]. In 2011, Kwak et al. published a TIRADS (Kwak-TIRADS) to detect suspicious malignant features: microcalcifications, solid composition, hypoechogenicity, a taller-than-wide shape, and an irregular/microlobulated margin [19]. In 2016, The Korean Thyroid Association/Korean Society of Thyroid Radiology (KTA/KSThR) proposed a pattern-based RSS (Korean-TIRADS) based on solidity and echogenicity with additional suspicious features (microcalcifications, non-parallel orientation, and spiculated/microlobulated margins) [20]. In 2015, The American Thyroid Association (ATA) announced a pattern-based, five-tier RSS with different risks of malignancy [21]. Similar to the Korean-TIRADS, the European Thyroid Association (ETA) in 2017 proposed a pattern-based five-tier RSS (EU-TIRADS) with US features showing a high probability of malignancy (irregular shape and margins, marked hypoechogenicity, solidity, and microcalcifications) [22]. Simultaneously, the American College of Radiology (ACR) published the scoring-based ACR TI-RADS [18].

Recently, several studies were carried out to compare the diagnostic performance of different US-based RSSs [13,14,17,23,24,25,26,27,28,29,30]. Although it is known that hyperfunctioning TNs have a very high probability of being benign and need no further diagnosis [31], none of these studies took the functional status of the TNs into account. Furthermore, in a previous study, our group demonstrated that a relevant proportion of hyperfunctioning TNs were classified as intermediate risk or high risk according to Kwak-TIRADS [32].

The aim of this study was to compare the diagnostic performance of five established US RSSs for non-autonomously functioning TNs in iodine deficiency.

## 2. Materials and Methods

### 2.1. Patients and Ethics

Since 2012, an increasing number of physicians specializing in thyroid diagnostics have been in constant communication regarding the diagnostic assessment of TNs, organized in the “German TIRADS Study Group” (GTSG). In recent years, seven institutions set up a continuously growing multicenter database containing the imaging and clinical data of over 2000 consecutive TNs. US features were recorded prospectively in real time immediately after the US examinations (see Section 2.2). Out of this pool, patients recorded between January 2012 and August 2020 were considered for the study. Their cases were consecutively recorded without influencing the treatment course, which was conducted according to guideline-based clinical decisions by the respective sites. Since August 2020, the rating of the RSSs was retrospectively conducted based on prospectively documented US features. Observers were blinded to the clinical results such as cytological and histopathological findings. Communication between the observers regarding difficult cases was, and is, consistently performed to reduce interobserver bias [33].

The inclusion criteria consisted of hypofunctioning or indifferent TNs on thyroid scintigraphy and the availability of cytological (FNC) or histopathological (surgery) diagnoses. Bethesda II lesions were considered benign. Scintigraphically hyperfunctioning TNs and those without scintigraphy as well as FNC findings outside Bethesda category II without histopathological evaluation were excluded. Scintigraphy scans were conducted according to the European guideline using 99 m-technetium-pertechnetate [31].

Recorded data comprised institution site, age, gender, number of TNs per patient, lesion size in three dimensions (crania–caudal, ventral–dorsal, medial–lateral), lesion functionality on scintigram, US features and RSS classifications (see Section 2.2), cytological findings according to the Bethesda System [34], and histopathological results.

The multicentric data collection was conducted according to the guidelines of the Declaration of Helsinki and approved by the Ethics Committee of the Medical Faculty of the University Hospital of Duisburg–Essen, Germany (ID: 16-7022-BO).

### 2.2. Ultrasound Examinations

US examinations were carried out according to the respective local standards with an emphasis on high-resolution, state-of-the-art image quality, and acquisition in transversal and sagittal orientation. Therefore, examination parameters, such as patient positioning, frequency, focus number and focus positioning, zoom, depth, gain, virtual convex mode, crossbeam mode, harmonic imaging modes, and breath-hold techniques were adapted to individual patient and nodule–specific requirements.

The following US devices were used:A Mindray DC-6 (Mindray Medical International Limited, Shenzhen, China) and Esaote MyLab 40 (Esaote SpA, Genova, Italy) equipped with a 10- and 12-MHz small parts probe;Hitachi EUB 5000 G (Hitachi Ltd., Chiyoda, Tokyo, Japan) equipped with a 5–10 MHz linear probe;Hitachi HI VISION Avius (Hitachi Ltd., Chiyoda, Tokyo, Japan) equipped with a 5–10 MHz linear probe; andGE LOGIQ E9 (GE Healthcare, Milwaukee, WI, USA) equipped with a 10–15 MHz linear probe.

The following US features were recorded:Composition: solid, <10, 10–50, 50–90, >90% cystic, spongiform;Echogenicity: (marked) hypoechoic, isoechoic, hyperechoic, completely cystic;Margin: sharp/smooth, macrolobulated, microlobulated, irregular, ill-defined, extrathyroidal extension (ETE);Calcifications/spots: none, colloidal-cystic associated spots, macrocalcifications, rim calcifications, rim calcifications with small extrusive soft tissue component (SESTC), microcalcifications; andShape: taller-than-wide (TTW), non-TTW, round.

Of these features, all TNs were classified according to the five RSSs: Kwak-TIRADS [19], ACR TI-RADS [18], EU-TIRADS [22], ATA Guidelines [21], and Korean-TIRADS [20].

### 2.3. Data Analyses and Statistics

Data were recorded on Excel software (Version 14.7.3, Microsoft Corporation, Redmond, WA, USA) and transferred to SPSS Statistics software (International Business Machines Corporation, Version 26.0, New York, NY, USA) for statistical analyses. Fisher’s exact test was conducted to evaluate group differences for ordinal values (e.g., US features). A Student’s *t* test was performed to investigate the differences among groups with normally distributed metric values (e.g., TSH-level, lesion size). For each RSS, calculations were made for positive predictive value (PPV), negative predictive value (NPV), sensitivity, specificity, diagnostic accuracy (ACC), positive likelihood ratio (LHR+), negative likelihood ratio (LHR-), diagnostic odds ratio (DOR), receiver operating curves (ROCs), and area under the curve (AUC). The AUC values were compared using a Hanley and McNeil test on MedCalc software (Version 20.009, Ostend, Belgium). If RSSs classifications were not applicable (N/A), the respective TN was not included in the analyses.

Cutoff values between benign and malignant for performance calculations were defined at 4c, TR5, 5, high, and high for Kwak-TIRADS, ACR TI-RADS, EU-TIRADS, Korean-TIRADS, and ATA Guidelines, respectively. For each test, *p* < 0.05 was considered significant.

## 3. Results

### 3.1. Patient Data and Clinical Characteristics of the Thyroid Nodules

A total of 1211 TNs in 849 patients (604 females, 71.1%; 249 males, 28.9%; aged 51 ± 14 years) were included in this study. The majority of the lesions were benign (*N* = 1022, 84.4%). Malignant lesions were diagnosed in 189 (15.6%) cases, of which 102 (54.0%) were carcinomas: papillary thyroid carcinoma (PTC) containing 19 (10.1%) papillary thyroid microcarcinomas (PTMC) and 43 (22.8%) follicular variants of PTC (FVPTCs), 10 (5.3%) follicular thyroid carcinomas (FTCs), 7 (3.7%) medullary thyroid carcinomas (MTCs), 5 (2.6%) poorly differentiated thyroid carcinomas (PDTCs), 1 (0.5%) anaplastic thyroid carcinoma (ATC), 1 (0.5%) metastasis of a colorectal cancer (CRC), and 1 (0.5%) manifestation of a Non-Hodgkin Lymphoma (NHL).

Histopathological and cytological results were available for 731 (60.4%) and 776 (64.1%) lesions, respectively. In total, 480 (39.6%) TNs were diagnosed as benign by cytology (Bethesda II) only. For 296 (24.4%) lesions, cytological and histopathological results were available. In 142 cases, Bethesda III/IV results were found on cytological examinations. The rate of malignancy in these TNs was 15.5% (Table 1).

The mean size (largest diameter) of the TNs was 26 ± 13 mm. Since in Germany thyroid scintigraphy is only regularly performed (irrespective of the TSH level) on TNs ≥ 10 mm, only eight (0.7%) TNs measured < 10 mm and 14 (1.1%) lesions showed a size of 10 mm. These were resected along with other lesions and their RSS classifications as well as scintigraphy findings were retrospectively assessed (with blinded histopathological results). The benign lesions were larger and more frequently hypofunctioning in the present study population (Table 2).

### 3.2. Ultrasound Features

US features that were documented for malignant and benign TNs are displayed in Table 3. Over 75% of the included carcinomas showed at least one of the following features: a solid composition, (marked) hypoechogenicity, and micro- or macrocalcifications, respectively. In contrast, over 75% of the benign lesions were characterized by sharp/smooth margins, non-TTW shape, missing calcifications, or demonstrating only colloidal-cystic associated spots. The sensitivity (specificity) of solid composition, hypochogenicity or marked hypoechogenicity, irregular or microlobulated shape, microcalcifications, and TTW for the detection of malignant TNs were 81.5% (47.6%), 84.7% (51.8%), 47.6% (92.2%), 55.0% (81.5%), and 33.3% (85.2%), respectively. The ACC values for solid components, (marked) hypoechogenicity, microlobulated or irregular margins, microcalcifications, and TTW were 52.8%, 56.9%, 85.3%, 77.3%, and 77.1%, respectively.

### 3.3. Risk Stratification Systems

All TNs were classifiable according to Kwak-TIRADS, ACR TI-RADS, and Korean-TIRADS. A total of 3 (0.2%, 1 malignant) and 135 (11.1%, 16 malignant) TNs could not be classified using EU-TIRADS and ATA Guidelines, respectively (Figure 1). The RSS classification results are displayed in Figure 2.

The PPV, NPV, Sensitivity, Specificity, and diagnostic accuracy ranged between 32.0% (EU-TIRADS) and 44.9% (Korean-TIRADS), 93.0% (ACR TI-RADS) and 95.6% (EU-TIRADS), 67.7% (ACR TI-RADS) and 83.5% (EU-TIRADS), 67.3% (EU-TIRADS) and 84.7% (Korean-TIRADS), and 69.8% (EU-TIRADS) and 82.0% (Korean-TIRADS), respectively (Table 4).

The ROCs of the investigated RSSs are shown in Figure 3. The AUC values were 0.803 (95% Confidence Intervals: 0.765–0.840), 0.795 (0.759–0.831), 0.800 (0.765–0.834), 0.805 (0.768–0.842), and 0.801 (0.765–0.837) for Kwak-TIRADS, ACR TI-RADS, EU-TIRADS, Korean-TIRADS, and ATA Guidelines, respectively. There were no differences in the AUC values (Table 5).

## 4. Discussion

One of the most dynamic fields in clinical thyroid research is the sonographic risk stratification of thyroid nodules. US devices are ubiquitous, and the procedure is a patient-friendly, cost-effective, and repeatable approach that has no side effects. Many different RSSs have been published in the recent years, and in the present study the diagnostic performances of five important ultrasound-based risk stratification systems (Kwak-TIRADS, ACR TI-RADS, EU-TIRADS, Korean-TIRADS, and ATA Guidelines) were evaluated in a population that has a high prevalence of TNs due to a long history of iodine deficiency [7,8].

Since 2012, the German TIRADS Study Group has been recording consecutive thyroid nodule cases from seven German institutions where there is a growing number of participating members. In this manner, a large database was built. Constant communication regarding difficult cases and the recent literature was conducted to achieve high performance levels in the application of RSSs and to reduce interobserver variability among the operators [33]. With the present multicenter trial, the group reported the first extensive German dataset regarding the diagnostic performance of five US-based RSSs for non-autonomous TNs.

Because the study focused on TNs that had been invasively diagnosed according to the clinical decision of the treating physicians, the preselected lesions (no hyperfunctioning TN, cytology or histopathology demanded) did not accurately represent the underlying patient population of Germany. Thus, malignant lesions were overrepresented: 15.5% in comparison to their natural incidence of <5% [35]. However, the data also contained TNs that had not been referred to the surgeons primarily for histopathological evaluation but had been resected as part of other surgical indications in multinodular goiters. This mitigates selection bias in favor of a higher classifications of the RSSs.

Meta-analyses are proposing sensitivities (specificities) for the detection of malignancy of 73–87% (53–56%), 63–78% (55–62%), 51–66% (79–83%), 40–54% (80–88%), and 27–53% (77–97%) for solid composition, hypoechogenicity, irregular margins, microcalcifications, and TTW shape, respectively [36,37,38].The calculated sensitivities and specificities for these US features in the current study were in good concordance with those in the literature. Diagnostic accuracies ranged between 52.8% (solid composition) and 85.3% (microlobulated or irregular margins).

The diagnostic accuracy of EU-TIRADS (69.8%) was inferior to that of Kwak-TIRADS (78.6%), ACR TI-RADS (77.9%), or Korean-TIRADS (82.0%), because of the relatively high number of EU5 classifications. ATA Guidelines showed a comparably high accuracy of 79.3% but a remarkable number of TNs (11.1%) were N/A. The ATA Guidelines provided an atlas that was primarily pattern-based, which was missing clear definition for isoechoic TNs with suspicious further US features. This problem has already been described in previous studies [33]. However, N/A TNs were excluded from the diagnostic performance calculations. Based on these results, Kwak-TIRADS, ACR TI-RADS, and Korean-TIRADS outperformed EU-TIRADS and ATA Guidelines in the study population, despite the AUC values on ROCs of all five RSSs being very similar (between 0.795 and 0.805) without significant differences (N/A TNs excluded). The diagnostic performance parameters were in concordance with the results of current meta-analyses (Table 6). Wei et al. reported a pooled sensitivity of 79% and a pooled specificity of 71% for mixed TIRADS studies. Pooled sensitivity (specificity) values of 98% (55%), 54–82% (53–90%), 66–74% (64–91%), 55–86% (28–95%), and 74–87% (31–88%) were published for Kwak-TIRADS, EU-TIRADS, ACR TI-RADS, Korean-TIRADS, and ATA guidelines, respectively. However, the cut-off values between benign and malignant lesions were partly different among the respective meta-analyses.

Considering the data from former iodine deficiency areas specifically, Dobruch-Sobczak et al. observed a sensitivity of 93.4% and a specificity of 54.6% for EU-TIRADS with a cut-off for EU5 in a Polish multicenter study containing 842 TNs (229 malignant) [44]. In a smaller study population from Austria (*N* = 195), EU-TIRADS, Kwak-TIRADS, ATA Guidelines, and French-TIRADS were assessed suitable for the differentiation between benign and malignant TNs. The authors found a sensitivity of 85% and a specificity of 45% with a cut-off of two or more positive US criteria. However, this was only true for the 45 included PTCs, but not for the eight FTCs [29]. In the present study, a large variety of different malignant lesions were observed, containing 54.0% PTC, 5.3% FTC, 3.7% MTC, 2.6% PDTC, 0.5% ATC, and 1% other cancer types. Therefore, to the best of our knowledge, the current data provide the most comprehensive results from an area with history of iodine deficiency. In a recently published Italian real-life setting study (single-center, retrospective, observational) that included 6474 cytologically investigated TNs and comprised five different RSSs, inferior sensitivities (50.1–94.5%), PPV (7.7–11.5%), and AUC values in ROC analyses (0.606–0.632) were reported [45]. Among other reasons, such as a different history of iodine supply between Germany and Italy [46], the superior performance of the RSSs in the current study may be due to the exclusion of non-autonomously functioning lesions. In a previous study, the GTSG revealed that a relevant number of hyperfunctioning TNs showed high-risk US patterns [32]. Scintigraphically guided preselection can therefore be recommended to improve the US-based risk stratification of TNs.

Further clinical examination data revealed larger sizes and a higher frequency of scintigraphically hypofunctioning lesions for benign compared to malignant TNs. However, since the decision for or against cytological or histopathological clarification of a TN was carried out as a comprehensive clinical decision, the data were affected by a selection bias after considering several additional findings such as laboratory results and disease-related symptoms. Therefore, over 80% of the lesions were hypofunctioning in the study population. The data showed a high sensitivity (75.1%) but a very low specificity (14.9%) for the hypofunctional feature for detecting malignant lesions. Due to this selection bias (especially the exclusion of hyperfunctioning lesions) these diagnostic parameters did not display the findings in a clinical routine. However, the majority of the malignant TNs showed up as hypofunctioning on scintigraphy scans, which was in accordance with the literature [47].

The multicentric study design allowed a patient enrolled in the study to be managed by different approaches during clinical practice. It needs to be underlined that this could have affected the results. Since only TNs that were characterized by scintigraphy were included, less than 1% of the TNs measured were < 10 mm. However, it is known that lesions < 10 mm can be detected as hyperfunctioning on scintigraphy and can be reliably assessed by I-124 positron emission tomography (PET)/US fusion imaging even in unfavorable localizations [47,48,49]. Furthermore, TIRADS have been proven to perform well in TNs < 10 mm [50].

So far, no uniform RSS has been established worldwide, although work has recently begun on a new international US-based RSS for TN. With the participation of several scientific societies, the so-called I-TIRADS will be proposed and established internationally as a uniform evidence-based system. Currently, different working groups are investigating individual ultrasound criteria [51]. In addition, promising data already exist regarding the use of artificial intelligence (AI) to identify ultrasound patterns. This technique could significantly reduce interobserver variability and account for regional differences such as site-typical normal findings via variable databases [52]. Another important pillar in the evaluation of TNs is related to the aforementioned topics: the establishment of (automated) structured reporting (SR). It is already well advanced in other diagnostic examination procedures such as mammography or prostate MRI as well as in professional study protocols [53,54]. Concepts for the implementation of AI pattern detection and SR in the field of thyroid US have already been proposed. In particular, the generation of automated findings from manually acquired ultrasound image data has the potential to provide considerable time savings for medical staff and may thus also have health and economic relevance for regions with a high prevalence of thyroid disease [55,56,57].

## 5. Conclusions

Kwak-TIRADS, ACR TI-RADS, Korean-TIRADS, and ATA Guidelines revealed high performance levels with diagnostic accuracies of about 80% and AUC values of approximately 0.8 without significant differences. However, over 10% of the TNs were not classifiable according to ATA Guidelines. The diagnostic performance of EU-TIRADS was slightly inferior in comparison with the aforementioned ultrasound risk stratification systems for thyroid nodules. Therefore, Kwak-TIRADS, ACR TI-RADS, and Korean-TIRADS can be preferentially recommended in areas with a history of iodine deficiency. Scintigraphic preselection to exclude hyperfunctioning nodules may improve the performance of ultrasound-based risk stratification systems.

## Figures and Tables

**Figure 1 cancers-13-04467-f001:**
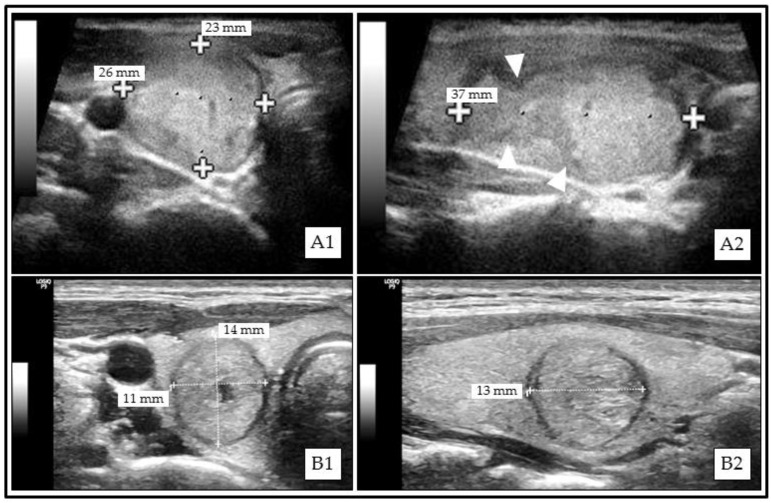
Examples of thyroid nodules (TNs) that could not be classified according to American Thyroid Association (ATA) Guidelines. (**A1**) (transversal)/(**A2**) (sagittal): Solid isoechoic papillary thyroid carcinoma (PTC) with irregular margins (**A2**, white triangle markers). (**B1**) (transversal)/(**B2**) (sagittal): Mainly solid isoechoic benign (Bethesda II) thyroid nodule (TN) with taller-than-wide (TTW) shape.

**Figure 2 cancers-13-04467-f002:**
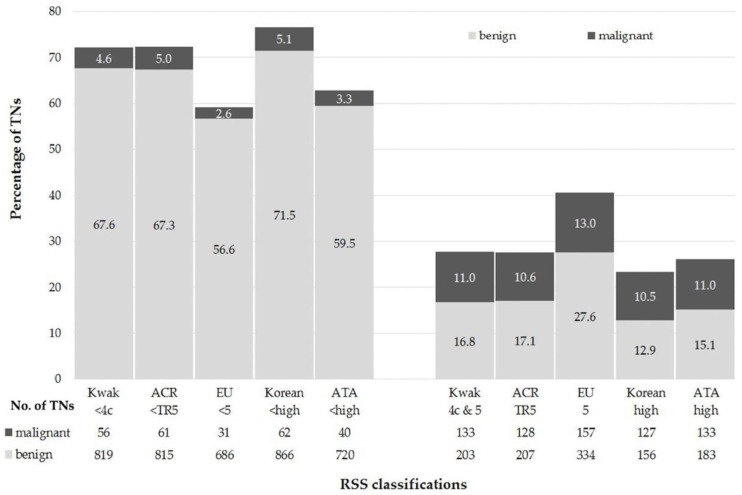
Performance of the risk stratification systems (RSSs). Abbreviations: TNs—Thyroid Nodules; ACR—American College of Radiology; EU—European Union; ATA—American Thyroid Association; RSS—Risk Stratification System.

**Figure 3 cancers-13-04467-f003:**
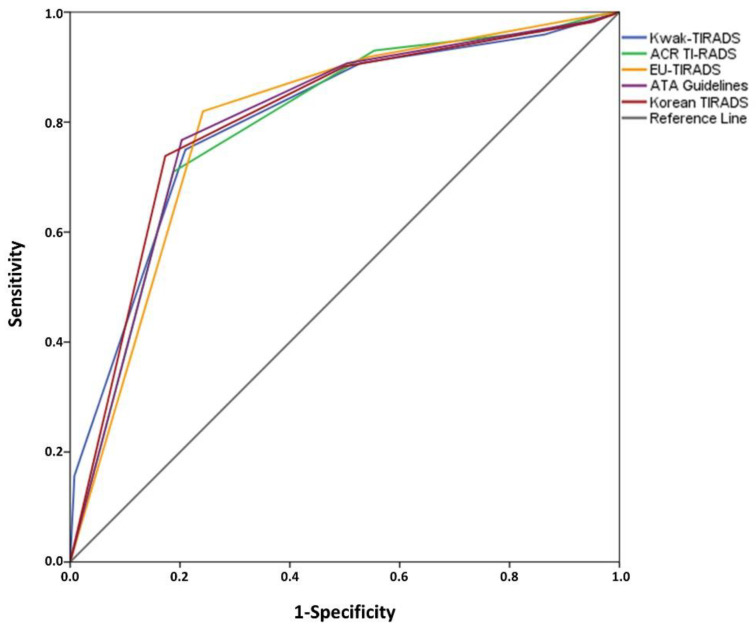
Receiver Operating Curves (ROCs) of the risk stratification systems (RSSs) *. * Thyroid nodules (TNs) that were not classifiable (N/A) are not included.

**Table 1 cancers-13-04467-t001:** Histopathological results of thyroid nodules (TNs) with fine-needle cytology (FNC) and surgery.

Bethesda Classifications [34]	All (*N* = 296) *N* (%)	Benign (*N* = 227) *N* (% of All)	Malignant (*N* = 69) *N* (% of All)
I—Nondiagnostic or Unsatisfactory	60 (20.3)	47 (78.3)	13 (21.7)
II—Benign	59 (19.9)	52 (88.1)	7 (11.9)
III/IV—AUS, FLUS, FN, suspicion for a FN	142 (48.0)	120 (84.5)	22 (15.5)
V—Suspicious for Malignancy	17 (5.7)	8 (47.1)	9 (52.9)
VI—Malignant	18 (6.1)	0 (0.0)	18 (100.0)

Abbreviations: AUS—Atypia of Undetermined Significance; FLUS—Follicular Lesion of Undetermined Significance; FN—Follicular Neoplasm.

**Table 2 cancers-13-04467-t002:** Scintigraphy results and lesion sizes.

Scintigraphy and Lesion Size	All (*N* = 1211) *N* (%)/Mean ± SD	Benign (*N* = 1022) *N* (%)/Mean ± SD	Malignant (*N* = 189) *N* (%)/Mean ± SD	*p*-Value
Scintigraphy	1211 (100.0)	1022 (100.0)	189 (100.0)	0.001
Indifferent	199 (16.4)	152 (14.9)	47 (24.9)	
Hypofunctioning	1012 (83.6)	870 (85.1)	142 (75.1)	
TN Size (mm)	26 ± 13	27 ± 13	19 ± 12	<0.001

Abbreviations: SD—Standard Deviation; TN—Thyroid Nodule.

**Table 3 cancers-13-04467-t003:** Ultrasound (US) features in relation to cytological and histopathological results.

US Features	All (*N* = 1211) *N* (%)	Benign (*N* = 1022) *N* (%)	Malignant (*N* = 189) *N* (%)	*p*-Value
Composition				
Solid	696 (57.5)	536 (52.4)	154 (81.5)	<0.001
<10% cystic	296 (24.4)	273 (26.7)	19 (10.1)	<0.001
10–50% cystic	160 (13.2)	148 (14.5)	12 (6.3)	0.002
50–90% cystic	27 (2.2)	24 (2.3)	3 (1.6)	0.788
>90% cystic	16 (1.3)	16 (1.6)	0 (0.0)	0.154
Spongiform	26 (2.1)	25 (2.4)	1 (0.5)	0.106
Echogenicity				
Hypo	530 (43.8)	419 (41.0)	102 (54.0)	<0.001
Marked hypo	132 (10.9)	74 (7.2)	58 (30.7)	<0.001
Iso	534 (44.1)	505 (49.4)	28 (14.8)	<0.001
Hyper	8 (0.7)	7 (0.7)	1 (0.5)	>0.999
Completely cystic	17 (1.4)	17 (1.7)	0 (0.0)	0.092
Margin				
Sharp/smooth	936 (77.3)	858 (84.0)	69 (36.5)	<0.001
Macrolobulated	43 (3.6)	40 (3.9)	2 (1.1)	0.05
Microlobulated	42 (3.4)	27 (2.6)	15 (7.9)	<0.001
Irregular	127 (10.5)	52 (5.1)	75 (39.7)	<0.001
Ill-defined	66 (5.5)	42 (4.1)	24 (12.7)	<0.001
ETE	7 (0.6)	3 (0.3)	4 (2.1)	<0.001
Calcifications				
None	742 (61.3)	660 (64.6)	75 (39.7)	<0.001
Colloidal	157 (13.0)	147 (14.4)	9 (4.8)	<0.001
Macro	155 (12.8)	102 (9.9)	42 (22.2)	<0.001
Rim	17 (1.4)	12 (1.2)	5 (2.6)	<0.001
Rim with SESTC	10 (0.8)	6 (0.6)	4 (2.1)	<0.001
Micro	289 (23.9)	189 (18.5)	104 (55.0)	<0.001
Shape				
TTW	214 (17.7)	151 (14.8)	63 (33.3)	<0.001
Non-TTW	980 (80.9)	857 (83.9)	113 (59.8)	<0.001
Round	27 (2.2)	14 (1.4)	13 (6.9)	<0.001

Abbreviations: US—Ultrasound; ACC—Diagnostic Accuracy; ETE—Extrathyroidal Extension; SESTC—Small Extrusive Soft Tissue Component; TTW—Taller Than Wide.

**Table 4 cancers-13-04467-t004:** Diagnostic performance parameters of the ultrasound risk stratification system (RSSs) for the differentiation between benign and malignant thyroid nodules (TNs).

Diagnostic Parameters	Kwak- TIRADS	ACR TI-RADS	EU- TIRADS	Korean- TIRADS	ATA Guidelines
Cut-off (benign vs. malignant)	4c	TR5	5	high	high
PPV	0.4	0.38	0.32	0.45	0.42
(CI-95)	(0.36–0.43)	(0.35–0.42)	(0.30–0.34)	(0.41–0.49)	(0.38–0.46)
NPV	0.94	0.93	0.96	0.93	0.95
(CI-95)	(0.92–0.95)	(0.92–0.94)	(0.94–0.97)	(0.92–0.94)	(0.93–0.96)
Sensitivity	0.7	0.68	0.84	0.67	0.77
(CI-95)	(0.64–0.76)	(0.61–0.74)	(0.78–0.88)	(0.60–0.73)	(0.70–0.83)
Specificity	0.8	0.8	0.67	0.85	0.8
(CI-95)	(0.78–0.82)	(0.77–0.82)	(0.64–0.70)	(0.82–0.87)	(0.77–0.82)
ACC	0.79	0.78	0.7	0.82	0.79
(CI-95)	(0.76–0.81)	(0.75–0.80)	(0.67–0.72)	(0.79–0.84)	(0.77–0.82)
LHR+	3.54	3.34	2.55	4.4	3.79
(CI-95)	(3.04–4.13)	(2.86–3.91)	(2.29–2.84)	(3.69–5.25)	(3.23–4.42)
LHR-	0.37	0.41	0.25	0.39	0.29
(CI-95)	(0.30–0.46)	(0.33–0.50)	(0.18–0.34)	(0.32–0.48)	(0.22–0.38)
DOR	9.58	8.26	10.4	11.37	13.08
(CI-95)	(6.78–13.57)	(5.88–11.62)	(6.93–15.62)	(8.03–16.11)	(8.87–19.30)

Abbreviations: RSS—Risk Stratification Systems; PPV—Positive Predictive Value; CI-95—95% Confidence Intervals; NPV—Negative Predictive Value; ACC—Diagnostic Accuracy; LHR+—Positive Likelihood ratio; LHR—Negative Likelihood ratio; DOR—Diagnostic Odds Ratio; TIRADS/TI-RADS—Thyroid Imaging and Reporting Data System; ATA—American Thyroid Association. Thyroid nodules (TNs) that were not classifiable (N/A) are not included.

**Table 5 cancers-13-04467-t005:** Comparison of Area under the Curve (AUC) values between the investigated risk stratification systems (RSSs) via Hanley and McNeil Test *.

RSSs	Kwak- TIRADS	ACR TI-RADS	EU- TIRADS	Korean- TIRADS	ATA Guidelines
Kwak-TIRADS	-	*p* = 0.760	*p* = 0.909	*p* = 0.941	*p* = 0.939
ACR TI-RADS	*p* = 0.760	-	*p* = 0.844	*p* = 0.702	*p* = 0.814
EU-TIRADS	*p* = 0.909	*p* = 0.844	-	*p* = 0.849	*p* = 0.969
Korean-TIRADS	*p* = 0.941	*p* = 0.702	*p* = 0.849	-	*p* = 0.879
ATA Guidelines	*p* = 0.939	*p* = 0.814	*p* = 0.969	*p* = 0.879	-

Abbreviations: RSS—Risk Stratification System; TIRADS/TI-RADS —Thyroid Imaging Reporting and Data System; ACR—American College of Radiology; EU—European Union; ATA—American Thyroid Association. * Thyroid nodules (TNs) that were not classifiable (N/A) are not included.

**Table 6 cancers-13-04467-t006:** Overview of meta-analyses regarding the diagnostic performance of ultrasound risk stratification systems (RSSs) for thyroid nodules (TNs).

Author, Year	No of Studies (TNs)	RSSs	Sensitivity Pooled (CI-95)	Specificity Pooled (CI-95)	LHR+ Pooled (CI-95)	LHR- Pooled (CI-95)	DOR Pooled (CI-95)	AUC on ROC
Wei et al., 2016 [39]	12 (10,437)	mixed	0.79	0.71	6.62	0.2	35.2	0.918
		TIRADS	(0.77–0.81)	(0.70–0.72)	(4.39–9.99)	(0.14–0.29)	(19.5–63.4)	
Migda et al., 2018 [40]	6 (10,926)	Kwak	0.98	0.55	2.67	0.05	51	0.938
			(0.98–0.99)	(0.54–0.56)	(1.69–4.20)	(0.04–0.07)	(15.2–170.8)	
Kim et al., 2020 [41]	29 (33,748)	ACR	0.66	0.91				0.89
			(0.56–0.75)	(0.87–0.94)				
		ATA	0.74	0.88				0.9
			(0.62–0.84)	(0.82–0.93)				
		Korean	0.55	0.95				0.88
			(0.38–0.70)	(0.90–0.98)				
		EU	0.82	0.9				0.91
			(0.71–0.89)	(0.77–0.96)				
Kim et al., 2020 [42]	34 (37,585)	ACR	0.7	0.89				
			(0.61–0.79)	(0.85–0.92)				
		Korean	0.64	0.93				
			(0.58–0.70)	(0.91–0.95)				
		EU	0.78	0.89				
			(0.64–0.88)	(0.77–95)				
Castellana et al., 2020 [43]	12 (18,750)	ACR	0.74	0.64	1.9	0.4	4.9	
			(0.61–0.83)	(0.56–0.70)	(1.6–2.3)	(0.3–0.6)	(3.1–7.7)	
		ATA	0.87	0.31	1.2	0.4	3.1	
			(0.75–0.94)	(0.24–0.40)	(1.0–1.4)	(0.2–0.7)	(1.3–7.1)	
		EU	0.54	0.53	1.4	0.6	2.2	
			(0.51–0.57)	(0.51–0.55)	(1.0–1.8)	(0.4–1.0)	(0.9–5.1)	
		Korean	0.86	0.28	1.2	0.5	2.5	
			(0.73–0.94)	(0.20–0.38)	(1.0–1.4)	(0.2–1.0)	(1.1–5.5)	

Abbreviations: RSSs—Risk Stratification Systems; TNs—Thyroid Nodules; LHR+—Positive Likelihood Ratio; LHR-—Negative Likelihood Ratio; DOR—Diagnostic Odds Ratio; AUC—Area Under The Curve; ROC—Receiver Operating Curves; CI-95—95% Confidence Intervals; TIRADS—Thyroid Imaging Reporting and Data System; ACR—American College of Radiology; EU—European Union; ATA—American Thyroid Association.

## Data Availability

The data presented in this study are openly available in FigShare at 10.6084/m9.figshare.14988171, reference number [58].

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
