# Peer review of "Diagnostic Performance of Kwak, EU, ACR, and Korean TIRADS as Well as ATA Guidelines for the Ultrasound Risk Stratification of Non-Autonomously Functioning Thyroid Nodules in a Region with Long History of Iodine Deficiency: A German Multicenter Trial"

_cancers, 2021, doi:10.3390/cancers13174467_

Round 1

Reviewer 1 Report

This article focused on ultrasound features and the resulting classifications of thyroid nodules (TNs) from an iodine deficient area using 5 risk stratification systems (RSSs) as Kwak-TIRADS, ACR TI-RADS, EU-TIRADS, Korean-TIRADS and ATA Guidelines. They conducted a retrospective multicenter study in German which enrolled 1211 TNs in 849 patients. The authors reported there was no significant differences between the RSSs according to ROC analysis. Besides, 11.1 % TNs could not be classified according to the ATA Guidelines. Otherwise, Kwak-TIRADS, ACR TI-RADS and Korean-TIRADS slightly outperformed EU-TIRADS and ATA Guidelines. This is a good opportunity to primarily recommend the preferable RSSs for areas of iodine deficiency. I have some questions for the authors about this article.

  1. The format of abstract should be modified for the abbreviations.
  2. Please clarify the study design in the title as long as in the methods.

  3. For the results of ultrasound features in Table 2 and Table could be simplified since there were no novel insights to present here.
  4. The imaging qualities of Figure 2 and Figure 3 could be justified.
  5. I am wondering the comparisons of diagnostic power of RSSs in the other areas also with iodine deficiency.
  6. Please express the shortage of this study.
  7. I cannot see the clear conclusion, which is the best RSS for German population, in the manuscript.

Reviewer 2 Report

The title reflects the subject of the study. This manuscript presents a clear and clinically useful message. It is well written in terms of clarity, style, and use of English language. Materials and methods are sufficiently detailed. The discussion section explains adequately the purpose of this study in the context of published information. The conclusions accurately and clearly explain the main results. The length of the manuscript is ideal. All figures are of good quality and relevant to the subject. All references are appropriate and current.

Reviewer 3 Report

I read with great interest the paper by Philipp Seifert , Simone Schenke , Michael Zimny , Alexander Stahl , Michael Grunert , Burkhard Klemenz , Martin Freesmeyer , Michael C. Kreissl , Ken Herrmann , Rainer Görges on the diagnostic performance of 5 common US-RSS for thyroid nodules.

Clearly, one of the strength sof this paper is that is is a contemporary report on real world data from larger number of secondary and tertiary thyroid centers. Moreover, since the authors a explicitly excluding hyperfunctioning nodules from their analysis, the diagnostic performance of the respective US-RSS has the potential to become more clinically meaningful.

One of the weaknesses is, that is is a retrolective analysis of prospectively documented clinical data and therefore, does not allow for an assessment of the procedurael diagnostic capacity of the US RSS – because of the systematic exclusion of BSRTC 2 cytologies and the incidental finding of malignant lesions during surgery intended to treat benign thyroid conditions. These adjustments tend to appreciably improve RSS diagnostic accuracy.

I would like to recommend the paper for publication, because it will add a relevant facette to the growing body of real life data on US-RSS. And there is a need for relevant practice reports in view of the attempts currently under way to discuss an “international TIRADS”.

I do have a number of comments thought to improve the manuscript.

M&M, Lines 94 and following: the definition of the cohort subselected from the 2000 in the database and entered into this analysis is not entirely clear to me. From the results (line 174) and discussion if would appear that tumors smaller than 10 mm were also excluded, as were hyperfunctional follicular adenomas (as determined by m99TECH), as well as those in whom no FNA or path report was available. I suggest to clarify the criteria and display the cohort and the excluded cases comparable to a CONSORT graph – and to present that graph as a supplementary file.

M&M line 135: It would appear to have been a retrolective analysis ? How was the RSS classification of individual nodules done ? They may have been recorded a-priori on site or ex-post by reclassification of recorded parameters. If indeed so, this should be stated and the rate of missing/incomplete values mentioned as well as measures to take account of interobserver bias – since all of the RSS were shown to be sensitive to bias on reclassification ?

Results

Line 206: When presenting the parameter diagnostic performance I would like to see the 95%Cis for at least PPV, NPV and Specificity as well as for the AUC. Notably, in a widespread disease such as thyroid nodules, NPV and sensitivity are the most powerful parameters to manage the rare malignant event (needing surgery) in the abundance of benign nodules (not requiring surgery).

Line 217: I suggest this table (5) to be moved into the supplemetary file section and to add the respective AUCs – if available.

Discussion

Line 260 -269 and 316-323 these are results and should be presents as such.

Table 4: in this table the authors present an PPV and a Sensitivity that is by for superior than in many other “real world data” report s (a.e. Petrone DOI10.1007/s00330-021-07703-5). I would speculate, that this imporved performance may be by virtue of excluding hyperfuncioning nodules. This is an original finding that I would very much like to encourage the authors to explore in more depth.

Reviewer 4 Report

Philipp Seifert et al. investigated the differences among various risk stratification methods by using ultrasound. They show that lower diagnostic accuracy of EU-TIRADS and ATA Guidelines than Kwak-TIRADS, ACR TI-RADS, and Korean-TIRADS depends on significant number of EU5 classification and N/A lesions, respectively. The authors imply that Kwak-TIRADS, ACR TI-RADS, and Korean-TIRADS have better performance comparing with other methods in the study group. The data supply higher sensitivity and specificity results for risk stratification systems comparing with previous studies that includes similar patient groups in iodine deficient areas. Additionally, the authors states that hyper-functioning thyroid nodules’ functionalities are also important for evaluation even though most are assessed as benign lesions.

This paper demonstrates that specific populations like iodine deficiency areas have different results of risk stratification methods. 

The authors relies on a single data set and to obtain meaningful results they'll need a  validation set. 

Regardless the clinical relevance is minimal and impact is probably negligible.

Round 2

Reviewer 1 Report

Thank you for the efforts in revisions. This manuscript is now worth to be published in this journal after remodeling of the figures (Fig 2 & 3) for clearer views.